# Inverted ILM flap technique versus conventional ILM peeling for idiopathic large macular holes: A meta-analysis of randomized controlled trials

**Guohai Chen**[1]◉, **Radouil Tzekov**[2]◉, **Fangzheng Jiang**[1], **Sihong Mao**[1], **Yuhua Tong**[1], **Wensheng Li**◉[3,4]*

1 Department of Ophthalmology, Quzhou People's Hospital, Quzhou, Zhejiang, PR China, 2 Department of Ophthalmology, University of South Florida, Tampa, Florida, United States of America, 3 Shanghai Aier Eye Hospital, Shanghai, PR China, 4 Aier School of Ophthalmology, Central South University, Changsha, Hunan Province, PR China

◉ These authors contributed equally to this work.
* drlws@qq.com

**Data Availability Statement:** All relevant data are within the manuscript and its Supporting Information files.

## Abstract

### Purpose

To compare the anatomical and visual outcomes of inverted internal limiting membrane (ILM) flap technique with the conventional ILM peeling for idiopathic large macular holes (MHs).

### Methods

A meta-analysis of randomized control trials (RCTs) using online databases including NCBI PubMed, ClinicalTrials.gov, and ISI Web of Science was performed. Anatomic success and type 1 closure rates, the mean postoperative best-corrected visual acuity (BCVA) and the mean change of BCVA from baseline were analyzed.

### Results

Out of 251 articles, four described clinical trials matching the inclusion criteria and were selected. They included 276 eyes (135 eyes in the inverted ILM flap group and 141 eyes in the ILM peeling group). All the studies used gas tamponade, with two studies having a follow-up duration of 3 months, while one study had a follow-up of 6 months and one study– 12 months. The meta-analysis demonstrated that anatomic success and type 1 closure rates (presence of neurosensory retina in MH) were better in the inverted ILM flap technique (odds ratio (OR) = 4.89; 95% confidence interval (CI), 2.09–11.47; P = 0.0003 and OR = 5.23; 95% CI, 2.83–9.66; P<0.00001). Similarly, the inverted flap technique was superior in terms of postoperative logMAR BCVA and mean change of logMAR BCVA from baseline (weighted mean difference (WMD) = 0.17, 95% CI, 0.11 to 0.24, P<0.00001 and WMD = 0.08, 95% CI, 0.01 to 0.16, P = 0.03)

**Funding:** Supported by the National Natural Science Foundation of China (No.81570875 to WS Li.), Welfare Technology Applied Research Program Fund of Science Technology Department of Zhejiang Province (No.LGF18H120002 to YH Tong.) and Quzhou Science and Technology Project (No.20172041 to GH Chen.)

**Competing interests:** The authors have declared that no competing interests exist.

## Conclusion

Inverted ILM flap treatment resulted in better closure rates and visual acuity when compared to the standard ILM peeling for large MHs.

## Introduction

Pars plana vitrectomy with internal limiting membrane (ILM) peeling and gas tamponade is currently considered the "gold standard" for macular hole (MH) surgery, with a promising success rate of more than 90% [1,2]. However, in cases of large MHs, with a diameter larger than 400 μm, the closure rate is lower, ranging from 56% to 85% [3,4]. Moreover, large MHs that have ended up achieving closure after conventional ILM peeling are more prone to display an "W-shape" pattern or a flat-open (lack of neurosensory retina in the hole) closure pattern, corresponding to a type 2 closure (according to Kang et al.) [5]. Although this outcome is considered an anatomic success, visual acuity is typically poorer compared to eyes with type 1 closure (presence of neurosensory retina in the hole), corresponding to a "U" or "V" pattern [5]. Thus, the treatment for large MHs could benefit from further improvements in surgical approach.

Michalewska et al. were the first to describe a technique for the treatment of idiopathic large MHs by creating an inverted ILM flap in 2010 [6]. In this method, the ILM is not completely removed from the retina but is left attached to the edges of the MH, and then inverted to cover the MH. Using this technique, those authors reported that the large MH closure rate was improved to 98% compared with 88% with the conventional ILM peeling and resulted in a better functional outcome [6].

Subsequently, several studies have suggested that the application of inverted ILM flap technique to treat large MH leads to a better visual acuity outcome [7–9]. However, other studies did not support this finding [10,11]. A systematic review and single-arm meta-analysis has focused on the inverted ILM flap technique for large MHs and reported that the anatomical closure and visual improvement rates were 95% and 75%, respectively [12]. However, most of the included studies were either retrospective in nature or lacked a control arm. A recent meta-analysis indicated that inverted ILM flap was recommended for large MHs [13]. This meta-analysis included both retrospective studies and randomized controlled trials (RCTs), and did not analyze the type 1 closure rates. We decided to conduct an independent assessment of the available literature data and to undertake a meta-analysis including only RCTs comparing the inverted ILM flap technique to the conventional ILM peeling for the treatment of idiopathic large MHs.

## Materials and methods

### Search strategy

We conducted searches of PubMed, ClinicalTrials.gov, and ISI Web of Science, using the terms ("macular hole") AND ("inner limiting membrane" OR "internal limiting membrane") AND "inverted"), with the language restricted to English, The final search was carried out on December 26, 2019. Additional search was carried out by exploring reference lists in the originally identified articles.

## Inclusion and exclusion criteria

The criteria we applied when published studies were considered eligible for this meta-analysis were: 1. study design: RCT, 2. population: participants had idiopathic MHs with a minimum MH diameter exceeding 400µm, 3. intervention: inverted ILM flap versus ILM peeling, and 4. outcome variables: a) proportion of cases with anatomic success and type 1 closure; b) post-operative best-corrected visual acuity (BCVA), and, c) post-operative change in BCVA. Articles reporting data from the same study, abstracts, letters to the editor, case reports, and review articles were excluded.

## Outcome measures

The primary outcome measure used in this analysis was an anatomic outcome, defined as the proportions of patients with anatomic success and type 1 closure after surgery. The secondary outcome measure was a visual function outcome, defined as the mean postoperative best-corrected visual acuity (BCVA) expressed as logarithm of the minimal angle of resolution (logMAR) and the mean change LogMAR BCVA from baseline.

## Data extraction

Two surgeons (G.C. and W.L.) reviewed all citations generated by the search and selected studies that matched the inclusion criteria, then extracted data from included studies. Uncertainty was resolved by discussion. The list of extracted items was as follows: first author's name, year of publication, country of origin, number of participants in each group, MH size, preoperative and postoperative BCVA, closure rates, dye used for ILM staining, type of tamponade and length of follow-up.

## Qualitative assessment

Two review authors (F.J. and S.M.) used a Jadad scale [14] to assess the methodologic qualities of RCTs. This system is a 5-point scale with 3 items: randomization, masking and participant withdrawals/dropouts. Each item is assigned 1 point when randomization is mentioned and 1 additional point when the randomization method was judged to be appropriate. Similarly, 1 point is assigned when masking is mentioned and 1 additional point is added when the method of masking was deemed appropriate. Studies assigned fewer than 3 points were judged to be of poor methodologic quality.

## Statistical analysis

The meta-analysis was conducted by using Cochrane Review Manager (RevMan, software version 5.1, Copenhagen, Denmark: The Nordic Cochrane Center, The Cochrane Collaboration, 2011). When analyzing continuous variables, the weighted mean difference (WMD) was calculated, while the odds ratios (OR) were obtained for dichotomous variables (e.g., number of eyes) and a 95% confidence interval (CI) was reported. $P<0.05$ was considered statistically significant on the test for overall effect. Inter-study heterogeneity was estimated by the Q statistic-test [15]. If the Q statistic test turned out as statistically significant ($P<0.05$), a random-effects model was used. In case where Q-statistic was not significant a fixed-effects model was applied. The level of bias in the selected publications was assessed by Begg's rank correlation test and by Egger's linear regression test with $P<0.05$ considered significant [16,17].

## Results

### Overall characteristics of selected trials and quality assessment

The search resulted in 251 articles being identified. Of these, 247 did not meet the inclusion criteria listed above and were rejected. This resulted in four studies remaining, which were included in this meta-analysis [6,18–20]. Fig 1 provides a flow diagram of the search procedure and results. In total, there were 276 eyes included in the meta-analysis: 135 eyes were included in the inverted ILM flap group, and 141 eyes were included in the ILM peeling group. All studies fulfilled the quality criteria (3 points or more on the Jadad scale). There was no statistically significant difference in BCVA of eyes assigned to the ILM flap group versus those assigned to the ILM peeling group in any included study at baseline. Three studies used Brilliant Blue G Ophthalmic Solution for ILM staining [18–20], while one study used Trypan Blue [6]. All the studies used gas tamponade at the end of surgery. Table 1 provides the characteristics and quality score of the included studies; all were RCTs, per our inclusion criteria.

### Anatomic outcome

Anatomic success after surgery was achieved in 128 of 135 eyes (94.8%) in the inverted ILM flap group compared with 111 of 141 eyes (78.7%) in the ILM peeling group. The pooled OR comparing inverted ILM flap technique to ILM peeling for the anatomic success rate were in favor of inverted ILM flap technique (OR = 4.89; 95% CI, 2.09–11.47; P = 0.0003), with no heterogeneity identified (P = 0.24) (Fig 2). In a further analysis, only the cases with type 1 closure were included. Type 1 closure after surgery was achieved in 113 of 135 eyes (83.7%) in the inverted ILM flap group compared with 75 of 141 eyes (53.2%) in the ILM peeling group. The pooled OR comparing inverted ILM flap technique to ILM peeling for the type 1 closure rate were also in favor of inverted ILM flap technique (OR = 5.23; 95% CI, 2.83–9.66; P<0.00001), with no heterogeneity identified (P = 0.13) (Fig 3).

### Visual function outcome

The combined results showed that the mean postoperative BCVA was significantly better for inverted ILM flap technique (0.59 logMAR) compared to ILM peeling (0.77 logMAR) (WMD = 0.17, 95% CI, 0.11 to 0.24, P<0.00001), with no heterogeneity identified (P = 0.45) (Fig 4). Three studies involving 175 eyes compared inverted ILM flap technique to ILM peeling in terms of the mean change in logMAR BCVA from baseline. The mean change in logMAR BCVA after surgery in inverted ILM flap technique group (0.29 logMAR) appeared to be better compared to ILM peeling group (0.21 logMAR) (WMD = 0.08, 95% CI, 0.01 to 0.16, P = 0.03), with no heterogeneity identified (P = 0.96) (Fig 5).

Begg's test and Egger's test indicated no statistically significant evidence of publication bias for any of the parameters.

## Discussion

In terms of anatomic outcomes, the results of our meta-analysis were in favor of inverted ILM flap technique. The MH closure was achieved in 94.8% and 78.7% in the inverted ILM flap group and the ILM peeling group, respectively. These rates of closure are similar to rates reported in the previous single-arm meta-analysis [12], despite using different studies, which strongly supports the conclusion that the inverted flap technique results in a better anatomical outcome. In our sub-analysis, the type 1 closure was achieved in 83.7% and 53.2% in the inverted ILM flap group vs. the ILM peeling group, respectively. In assessing the mean postoperative logMAR BCVA and the mean change in logMAR BCVA from baseline, the results

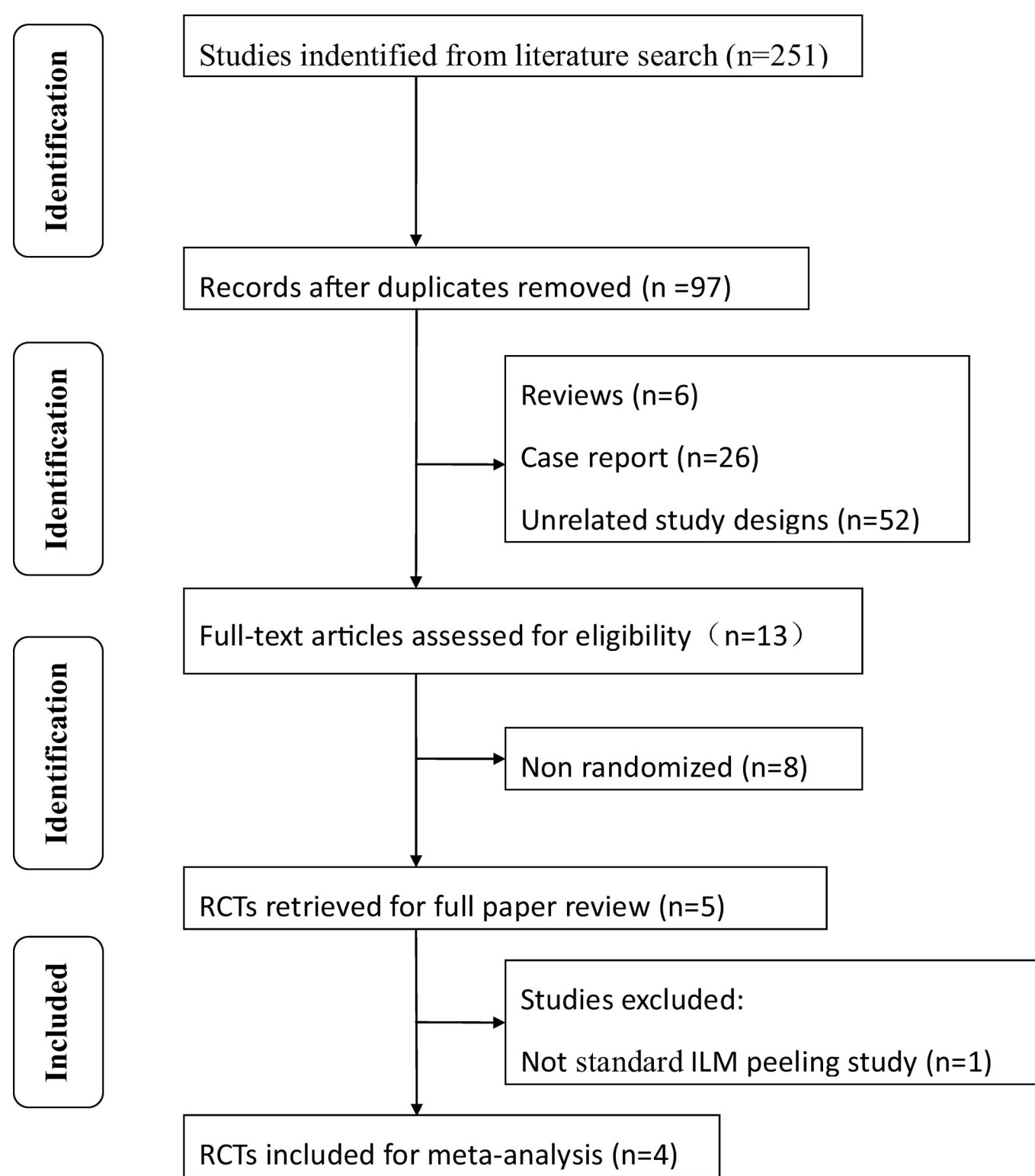

**Fig 1. Flow diagram of studies included in this meta-analysis.** RCT, randomized controlled trial.

**Table 1. Characteristics and quality scores of included studies.**

| Study group (year) | Design | Location | ILM staining | Tamponade | Follow-up (months) | Assigned method | No. eyes | Mean age of participants (year) | Mean minimum MH diameter (μm) | Mean baseline BCVA (LogMAR) | Jadad score |
|---|---|---|---|---|---|---|---|---|---|---|---|
| Michalewska (2010) [6] | RCT | Poland | Trypan blue | Air | 12 | Inverted ILM Flap | 50 | 66.0 | 698 ± 108 | 1.10 | 3 |
| | | | | | | ILM Peeling | 51 | 65.0 | 759 ± 300 | 0.92 | |
| Kannan (2018) [18] | RCT | India | BBG | SF$_6$ | 6 | Inverted ILM Flap | 30 | 59.4 | 803 ± 120 | 0.75 ± 0.22 | 3 |
| | | | | | | ILM Peeling | 30 | 61.2 | 759 ± 85 | 0.79 ± 0.24 | |
| Manasa (2018) [19] | RCT | India | ILM-Blue | SF$_6$ | 3 | Inverted ILM Flap | 43 | 63.4 | 673 | 0.99 ± 0.25 | 3 |
| | | | | | | ILM Peeling | 48 | 60.9 | 657 | 1.10 ± 0.28 | |
| Velez-Montoya (2018) [20] | RCT | Italy | BBG | SF$_6$ or C$_3$F$_8$ | 3 | Inverted ILM Flap | 12 | 64.2 | 608 ± 213 | 0.95 ± 0.20 | 4 |
| | | | | | | ILM Peeling | 12 | 61.8 | 522 ± 82 | 0.93 ± 0.50 | |

Abbreviations: RCT—randomized controlled trial; ILM—internal limiting membrane; BBG—Brilliant Blue G Ophthalmic Solution; ILM-Blue—another name for Brilliant Blue G Ophthalmic Solution in markets outside the USA; SF6,—sulfur hexafluoride; C$_3$F8,—octafuoropropane; MH—macular hole; BCVA—best corrected visual acuity; LogMAR—logarithm of the minimal angle of resolution.

were also in favor of inverted ILM flap technique. Moreover, the difference in terms of the mean postoperative logMAR BCVA (0.17 logMAR) between these two groups exceeded 0.1 logMAR (equivalent to one line on the ETDRS chart), suggestive of clinically significant difference. However, a subset of studies [18–20] reporting change from preoperative BCVA to postoperative BCVA indicates a difference of only 0.08 logMAR, which underscores the needs for more studies with more detailed reporting of vision function outcome.

The mechanism by which the inverted ILM flap technique works is not fully understood. Histopathologic studies confirmed that ILM being a base membrane acts as a scaffold for gliosis to close large MHs [9]. Besides gliosis, the ILM flap seals the MH by secluding communication between the vitreous and subretinal space, creating a closed compartment enabling the RPE to pump out fluid effectively, preventing further seepage of fluid and, hence, keeping the hole dry [19]. Shiode et al. experimentally proved that some of the constituents of ILM enhance the proliferation and migration of Müller cells in vitro [21]. Moreover, neurotrophic factors and basic

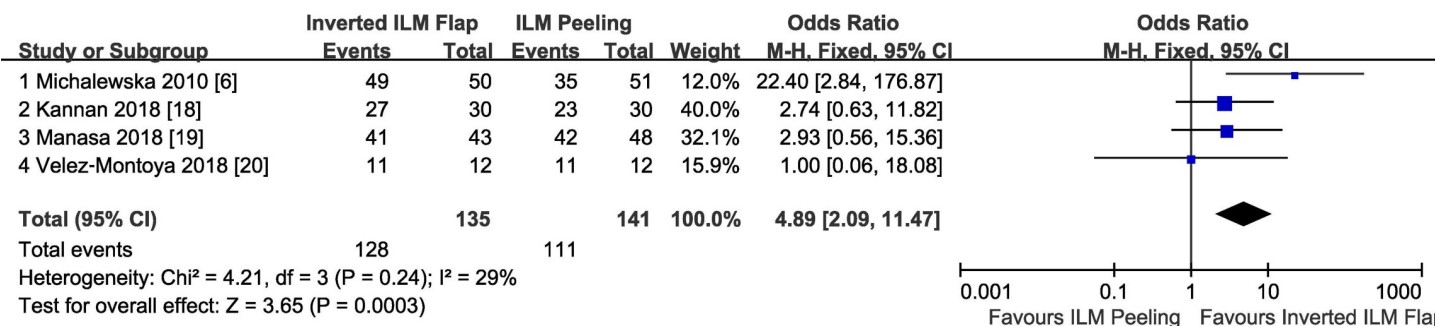

**Fig 2. Overall MH closure rate after surgery comparing inverted ILM flap with conventional ILM peeling.** ILM, internal limiting membrane; M-H, Mantel-Haenszel; CI, confidence interval.

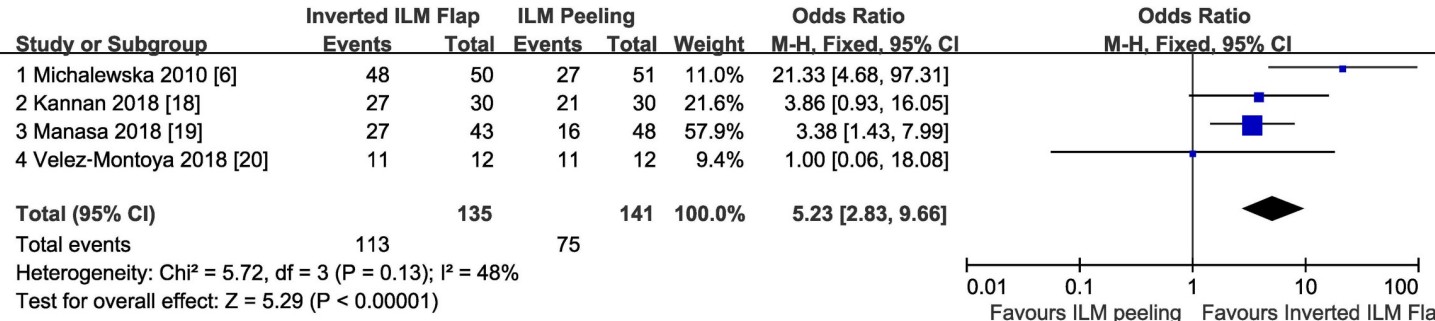

**Fig 3. Type 1 closure rate after surgery comparing inverted ILM flap with conventional ILM peeling.** ILM, internal limiting membrane; M-H, Mantel-Haenszel; CI, confidence interval.

fibroblast growth factor (bFGF) retained on the surface of the ILM flap and secreted by the migrating Müller cells could stimulate the survival of retinal cells [21]. Thus, it is likely that Müller cell gliosis, and humoral factors could contribute to the closure of large MHs.

Imai et al. proposed OCT-based evaluation of MH closure following vitrectomy [22]. Briefly, U-type closure (normal foveal contour) is the most favorable type of closure outcome. Next, V-type (steep foveal contour) shows notch configuration covered with thin retinal tissue, but is considered less favorable. Finally, W-type (foveal defect of neurosensory retina) is not a favorable type closure. Because the distinction between the "U" and "V" pattern was sometimes unclear, Kang et al. categorized macular closure types into two patterns: type 1 closure (closed without foveal neurosensory retinal defect), corresponding to the "U" and "V" pattern, and type 2 closure (closed with foveal neurosensory retinal defect), corresponding to the "W" pattern [5]. The magnitude of visual improvement in the postoperative period was greater in type 1 closures compared to type 2 closures [5]. This could be explained by the hypothesis that larger amounts of residual neurosensory retina are correlated with better visual function. Postoperative examination of the fovea indicated that patients with U-type closure had better photoreceptor layer appearance and normal retinal thickness at the end of the follow-up [23]. Thus, the aim is to obtain a type 1 anatomical closure in large MHs, especially the U-type closure, to improve functional outcomes. However, it is difficult to close large retina defects by glial tissue. Hence, large MHs have a tendency to remain open or to close in an unphysiological type 2 pattern [5,24]. Our meta-analysis indicates that inverted ILM flap technique achieved higher type 1 closure rate compared to conventional ILM peeling and the visual outcomes were also significantly better in the inverted ILM flap group. This finding indicates that the high anatomical success rate of this technique also resulted in postoperative functional improvement.

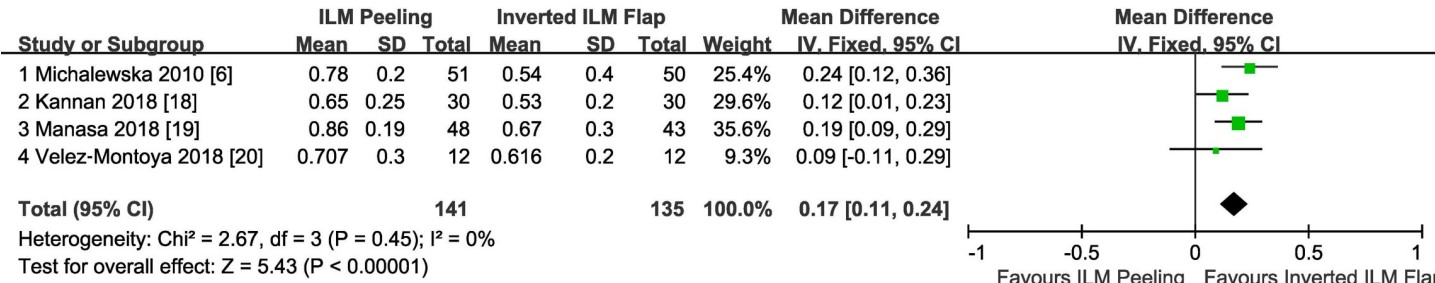

**Fig 4. Mean postoperative best corrected visual acuity (logMAR units) comparing inverted ILM flap technique with ILM peeling.** ILM, internal limiting membrane; SD, standard deviation; IV, inverse variance; CI, confidence interval.

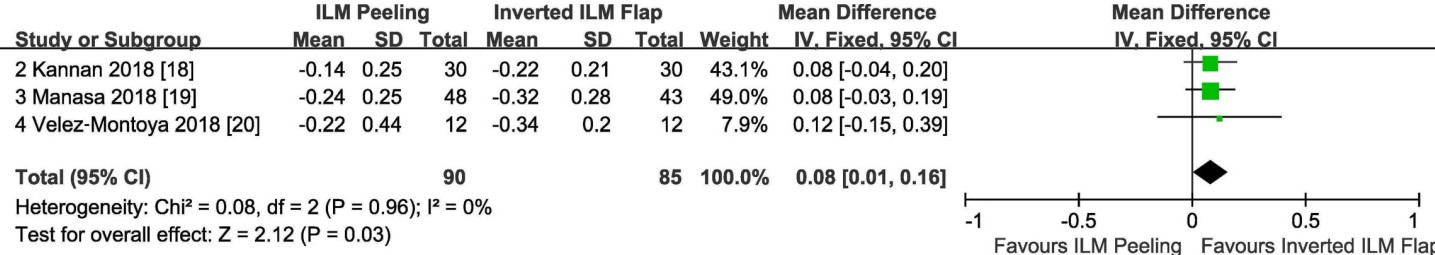

**Fig 5. Mean change in best corrected visual acuity (logMAR units) from preoperative baseline comparing inverted ILM flap technique with ILM peeling.** ILM, internal limiting membrane; SD, standard deviation; IV, inverse variance; CI, confidence interval.

This work carries some limitations which should be acknowledged. One of the main limitations is that all included studies were carried out with small or very small sample sizes. However, as large macular hole is an uncommon condition, it is difficult to come up with a large sample size operated by a single surgeon in a limited time period. Second, a potential source of heterogeneity is different trial duration. Thus, two out of the four studies had a follow-up duration of 3 months. After a successful closing of the MH, the restoration of the retina may take time and visual acuity is usually better after a long-term observation than in short postoperative time [25]. RCTs of longer duration (at least 12 months) would be preferable to assess more definitively long-term efficacy. Third, different gas tamponade may influence the results. However, several studies found no significant difference in terms of anatomical closure rate, final visual acuity improvement between longer acting gas (octafluoropropane) and short acting gas (sulfur hexafluoride or room air) in cases of MHs with the follow-up duration of 3–12 months [26–29]. Fourth, one study (Manasa et al.) did not provide a standard deviation for MHs in baseline, but the minimum MH diameter exceeding 600μm [19], which fulfilled our inclusion criteria. Finally, in this meta-analysis, only BCVA was used as a sole indicator for vision function outcome. BCVA is a subjective measurement and determines the visual function only of small retinal area under optimal conditions. Other visual function methods, like multifocal electroretinogram (mfERG), which is an objective measurement and provides a topographic electrophysiological mapping of the central retina may be more suitable to provide a more comprehensive evaluation of the central visual function. Recently, one study using mfERG demonstrated differences in the bioelectrical response between the lower part of the retina containing an ILM flap and a symmetrical region of the upper retina without the ILM flap. This could indicate a very limited effect on retinal function by the inverted flap [30].

In summary, the present meta-analysis confirmed the theoretical advantages of the inverted ILM flap technique over conventional ILM peeling for idiopathic large MHs in terms of better anatomic outcome and visual outcome. Although based on a limited number of studies, this finding supports the notion that the inverted ILM flap is an effective technique for the treatment of idiopathic large MHs.

## Supporting information

**S1 Checklist.**
(DOC)

## Author Contributions

**Conceptualization:** Radouil Tzekov, Wensheng Li.

**Data curation:** Guohai Chen, Wensheng Li.

**Formal analysis:** Guohai Chen, Radouil Tzekov.

**Funding acquisition:** Guohai Chen, Yuhua Tong, Wensheng Li.

**Methodology:** Guohai Chen, Radouil Tzekov, Fangzheng Jiang, Sihong Mao, Wensheng Li.

**Software:** Guohai Chen.

**Supervision:** Fangzheng Jiang.

**Validation:** Fangzheng Jiang, Sihong Mao, Yuhua Tong.

**Visualization:** Yuhua Tong.

**Writing – original draft:** Guohai Chen.

**Writing – review & editing:** Radouil Tzekov, Fangzheng Jiang, Sihong Mao, Yuhua Tong, Wensheng Li.

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
