## [Decision Letter · Decision Letter 0]

5 May 2020

PONE-D-20-05200

Inverted ILM flap technique versus conventional ILM peeling for idiopathic large macular holes: a meta-analysis of randomized controlled trials

PLOS ONE

Dear Dr Li,

Thank you for submitting your manuscript to PLOS ONE. After careful consideration, we feel that it has merit but does not fully meet PLOS ONE’s publication criteria as it currently stands. Therefore, we invite you to submit a revised version of the manuscript that addresses the points raised during the review process.

ACADEMIC EDITOR: Please  reply to comments by reviewer point by point

We would appreciate receiving your revised manuscript by Jun 19 2020 11:59PM. To enhance the reproducibility of your results, we recommend that if applicable you deposit your laboratory protocols in protocols.io, where a protocol can be assigned its own identifier (DOI) such that it can be cited independently in the future. For instructions see: http://journals.plos.org/plosone/s/submission-guidelines#loc-laboratory-protocols

We look forward to receiving your revised manuscript.

Kind regards,

Shree K. Kurup, MD

Academic Editor

PLOS ONE

Journal Requirements:

2. Thank you for stating the following financial disclosure:""The funders had no role in study design, data collection and analysis, decision to publish, or preparation of the manuscript.""

Please provide an amended Funding Statement that declares *all* the funding or sources of support received during this specific study (whether external or internal to your organization) as detailed online in our guide for authors at http://journals.plos.org/plosone/s/submit-now.  

Additional Editor Comments (if provided):

please review comments and reply point by point. This MS has intrinsic challenges base don methodology

Reviewers' comments:

Reviewer's Responses to Questions

**Comments to the Author**

1. Is the manuscript technically sound, and do the data support the conclusions?

Reviewer #1: Yes

2. Has the statistical analysis been performed appropriately and rigorously? 

Reviewer #1: Yes

3. Have the authors made all data underlying the findings in their manuscript fully available?

Reviewer #1: Yes

4. Is the manuscript presented in an intelligible fashion and written in standard English?

Reviewer #1: Yes

5. Review Comments to the Author

Reviewer #1: The meta-analysis entitled „Inverted ILM flap technique versus conventional ILM peeling for idiopathic large macular holes: a meta-analysis of randomized controlled trials“ is well-conducted the results sound, the information interesting at a first glance. There are, nevertheless, some aspects deserving closer attention:

Methodologically, a follow up time of 3 months limits the functional meaning of data, so that functional 3-month data should probably be censored. If not, this has to be pointed on in abstract and text.

The gas tamponade has been discussed but not mentioned in the abstract (i.e. no oil) whereas the dye used for staining was not mentioned though possibly a confounder of results (the toxic ICG is bound in the ILM for long time, whereas trypan blue is not.

Abstract: explain or remove type 1 closure rates; explain WMD

Introduction: Remove redundant “however” and improve wording, namely of the last paragraph, lines 57-63.

M&M: include dye used for staining and specify tamponade, both probably in a table which also includes population demographics to help understanding differences.

Confirm, that MH closure types were reported by all 4 studies included here or re-consider primary outcome definition.

Line 88 and lines 115-7, 187: improve wording.

Lines 134-5, 146-7, 162-3: remove, reiterations.

Line 148: remove “with statistically significant” redundancy

Line 172: change to …closure of large MHs

Line 214: replace conformed by confirmed

Table 1: add reference number. Manasa: study quality supportable if standard deviation was not determined? Overall rating for Velez-Montoya low because of a small sample size and a short follow up. To be mentioned, data to be used with specific care.

Figures 2-5: Same order of studies as in the table to facilitate reading. Was the sample size respected for calculation of the OR? Use a decimal or logarithmic scale throughout facilitates reading of the graph.

6. PLOS authors have the option to publish the peer review history of their article (what does this mean?). If published, this will include your full peer review and any attached files.

Reviewer #1: No

---

## [Author Response · Author response to Decision Letter 0]

2 Jun 2020

Reviewer #1: The meta-analysis entitled „Inverted ILM flap technique versus conventional ILM peeling for idiopathic large macular holes: a meta-analysis of randomized controlled trials“ is well-conducted the results sound, the information interesting at a first glance. There are, nevertheless, some aspects deserving closer attention:

1.Methodologically, a follow up time of 3 months limits the functional meaning of data, so that functional 3-month data should probably be censored. If not, this has to be pointed on in abstract and text.

Response: Two out of the four studies used in the analysis had a follow-up duration of 3 months. Censoring it would incapacitate this meta-analysis. However, we completely agree with the reviewer that this represents a major limitation and should be clearly communicated to the reader. Therefore, we have added this in the revised version of abstract, in addition to this being pointed out as one of the major limitations in the text (Discussion).

2.The gas tamponade has been discussed but not mentioned in the abstract (i.e. no oil) whereas the dye used for staining was not mentioned though possibly a confounder of results (the toxic ICG is bound in the ILM for long time, whereas trypan blue is not.

Response: Thank you for this valuable suggestion. We have revised the abstract accordingly and the gas tamponade is now mentioned in the text. We agree that ICG could be bound to ILM for a long time and be toxic to the surrounding tissue. None of the studies selected for this meta-analysis have used ICG; 2 studies used Brilliant Blue G, one used Trypan Blue and one used ILM Blue (just another name for Brilliant Blue G Ophthalmic Solution in other markets outside the USA). The dye used for staining has been mentioned in the revised version of text. We have also added the dye used for staining and specified the type of tamponade in the revised version of Table 1.

3. Abstract: explain or remove type 1 closure rates; explain WMD.

Response: We have explained type 1 closure rates and WMD in the revised version of abstract.

4. Introduction: Remove redundant “however” and improve wording, namely of the last paragraph, lines 57-63.

Response: We have removed the redundant “however” and reworded this paragraph in the revised version of the text.

5. M&M: include dye used for staining and specify tamponade, both probably in a table which also includes population demographics to help understanding differences.

Response: We have added the dye used for staining and specify tamponade in the revised version of Table 1. This information was also added in the revised text of the Materials and Methods.

6. Confirm, that MH closure types were reported by all 4 studies included here or re-consider primary outcome definition.

Response: We have revised the primary outcome definition to “anatomic outcome”, and the secondary outcome definition to “visual function outcome”

7. Line 88 and lines 115-7, 187: improve wording.

Response: We have reworded these sentences in the revised version of the text.

8. Lines 134-5, 146-7, 162-3: remove, reiterations.

Response: We have removed these sentences in the revised version of the text. 

9. Line 148: remove “with statistically significant” redundancy.

Response: We agree with the reviewers’ comment and have removed the word in the revised version of the text. 

10. Line 172: change to …closure of large MHs.

Response: Thank you for pointing this out this discrepancy. We have changed it in the revised version of the manuscript.

11. Line 214: replace conformed by confirmed.

Response: Thank you for pointing out this typo. We have corrected it.

12.Table 1: add reference number. 

Response: We have added the reference numbers in the revised version of Table 1. 

Manasa: study quality supportable if standard deviation was not determined?

Response:The standard deviation of macular hole diameter was not available in Manasa (2018) study. This limitation was pointed out in the revised text of the Discussion.

Overall rating for Velez-Montoya low because of a small sample size and a short follow up. To be mentioned, data to be used with specific care.

Response: We used the Jadad scale to assess the methodologic qualities of RCTs, using 3 criteria: randomization, masking and participant withdrawals/dropouts. The sample size and follow-up time were not included as criteria in the Jadad scale; therefore Velez-Montoya (2018) ended up with 4 points on the Jadad scale. On the other hand, we completely agree with the reviewer that both small sample size and short length of follow-up are major limitations to this (and similar analyses) and have pointed out these limitations in the relevant paragraph in Discussion. 

13. Figures 2-5: Same order of studies as in the table to facilitate reading. 

Response: We have reordered the studies in the Figures to match the order in Table 1.

Was the sample size respected for calculation of the OR? Use a decimal or logarithmic scale throughout facilitates reading of the graph.

Response: The sample size is accounted for in the calculation of the OR. The models used to generate the plots in Figure 2 and 3 used the Odds Ratio, while the models used to generate the plots in Figure 4 and 5 used the weighted mean difference. Therebefore, the scales for these figures have to be different.

---

## [Decision Letter · Decision Letter 1]

8 Jul 2020

Inverted ILM flap technique versus conventional ILM peeling for idiopathic large macular holes: a meta-analysis of randomized controlled trials

PONE-D-20-05200R1

Dear Dr. Li,

We’re pleased to inform you that your manuscript has been judged scientifically suitable for publication and will be formally accepted for publication once it meets all outstanding technical requirements.

Kind regards,

Shree K. Kurup, MD

Academic Editor

PLOS ONE

Additional Editor Comments (optional):

This MS is now suitable for consideration

Reviewers' comments:

Reviewer's Responses to Questions

**Comments to the Author**

1. If the authors have adequately addressed your comments raised in a previous round of review and you feel that this manuscript is now acceptable for publication, you may indicate that here to bypass the “Comments to the Author” section, enter your conflict of interest statement in the “Confidential to Editor” section, and submit your "Accept" recommendation.

Reviewer #1: All comments have been addressed

2. Is the manuscript technically sound, and do the data support the conclusions?

Reviewer #1: Yes

3. Has the statistical analysis been performed appropriately and rigorously? 

Reviewer #1: Yes

4. Have the authors made all data underlying the findings in their manuscript fully available?

Reviewer #1: Yes

5. Is the manuscript presented in an intelligible fashion and written in standard English?

Reviewer #1: Yes

6. Review Comments to the Author

Reviewer #1: After careful revision respecting the reviewer critics, the manuscript now reads interesting and seem acceptable

7. PLOS authors have the option to publish the peer review history of their article (what does this mean?). If published, this will include your full peer review and any attached files.

Reviewer #1: **Yes: **Justus G. Garweg

---

## [Editor Report · Acceptance letter]

14 Jul 2020

PONE-D-20-05200R1 

Inverted ILM flap technique versus conventional ILM peeling for idiopathic large macular holes: a meta-analysis of randomized controlled trials 

Dear Dr. Li:

I'm pleased to inform you that your manuscript has been deemed suitable for publication in PLOS ONE. Congratulations! Your manuscript is now with our production department. 

Kind regards, 

on behalf of

Dr. Shree K. Kurup 

Academic Editor

PLOS ONE